# A sustainable ultra-high strength Fe18Mn3Ti maraging steel through controlled solute segregation and α-Mn nanoprecipitation

A. Kwiatkowski da Silva [1✉], I. R. Souza Filho [1], W. Lu[1,2], K. D. Zilnyk [3], M. F. Hupalo [4], L. M. Alves [4], D. Ponge[1], B. Gault [1,5] & D. Raabe[1]

The enormous magnitude of 2 billion tons of alloys produced per year demands a change in design philosophy to make materials environmentally, economically, and socially more sustainable. This disqualifies the use of critical elements that are rare or have questionable origin. Amongst the major alloy strengthening mechanisms, a high-dispersion of second-phase precipitates with sizes in the nanometre range is particularly effective for achieving ultra-high strength. Here, we propose an alternative segregation-based strategy for sustainable steels, free of critical elements, which are rendered ultrastrong by second-phase nano-precipitation. We increase the Mn-content in a supersaturated, metastable Fe-Mn solid solution to trigger compositional fluctuations and nano-segregation in the bulk. These fluctuations act as precursors for the nucleation of an unexpected α-Mn phase, which impedes dislocation motion, thus enabling precipitation strengthening. Our steel outperforms most common commercial alloys, yet it is free of critical elements, making it a new platform for sustainable alloy design.

[1] Department of Microstructure Physics and Alloy Design, Max-Planck-Institut für Eisenforschung, Düsseldorf, Germany. [2] Department of Mechanical and Energy Engineering, Southern University of Science and Technology, Shenzhen, PR China. [3] Department of Materials & Processing Technology. Instituto Tecnológico de Aeronáutica (ITA), São José dos Campos, Brazil. [4] Department of Materials Engineering, Universidade Estadual de Ponta Grossa, Ponta Grossa, Brazil. [5] Department of Materials, Royal School of Mines, Imperial College London, London, UK. ✉email: a.kwdasilva@mpie.de

Steels are all around, enabling multiple key technologies—from swords to steam engines, automobiles, bridges, skyscrapers, wind mills and screws in our flat-packed furniture. Steels are the most widely used structural metallic alloys, and comprise an area of intense research and development, with many new variants designed every year, sometimes from the nanoscale up[1]. While recent trends in metallurgy aim at realizing new mechanical properties through highly alloyed compositional tuning[2,3], sustainability and social responsibility goals encourage us instead to use lean compositions and nanostructure tuning[4]. Along such lines, medium manganese steels have emerged as an attractive high-strength alloy class[5–7], leaning on the Earth abundant Mn as major alloying element. These steels are typically produced via martensitic transformation, by quenching a face-centered cubic (FCC) austenite (γ) high temperature phase into a supersaturated body-centered cubic (BCC) ferritic (α) phase. Figure 1a shows the iron-manganese binary phase diagram. The metastable Mn-rich martensitic phase is subsequently annealed to trigger austenite nucleation, preceded by the segregation (adsorption) of Mn to the numerous grain boundaries and dislocations inside the martensite[8,9]. The strong tendency for segregation in this system is connected to the ferromagnetism of Fe and the anti-ferromagnetism of Mn. Figure 1b shows a metastable Fe-Mn phase diagram calculated using only the α phase. The mixture of Fe and Mn tends to phase separate in two phases, $\alpha_1$ (Fe-rich, ferromagnetic) and $\alpha_2$ (Mn-rich, paramagnetic). Since most Fe-Mn alloys are relatively dilute with a global content between 4 and 12 wt.%, this segregation occurs typically at grain boundaries, eventually leading to the heterogeneous nucleation of a second phase[8,9].

Here we design a compositionally lean Fe18Mn3Ti (wt%) ultra-high strength steel, susceptible to homogeneous phase decomposition, assisted by segregation and composed of the three most abundant transition metals in Earth's crust. This alloy composition is designed to be unstable against compositional fluctuations at the intended aging temperature, around 450 °C (see "Methods" for details). These fluctuations act as precursors for the nucleation of an unexpected α-Mn nano-precipitate phase, which can decrease the mobility of the dislocations in the matrix, enabling precipitation strengthening of the martensitic matrix. We add 3 wt.% Ti to allow the transformation of austenite to α-martensite during quenching and cold rolling[10], preventing a high fraction of retained austenite and epsilon martensite (with hexagonal lattice structure), and stabilize the second phase α-Mn precipitates[11]. Such strategy avoids the addition of critical elements such as Co and Mo which enable intermetallic precipitation in conventional ultra-high strength maraging steels. Ni is also completely replaced by Mn which participates both in austenite stabilization and precipitate formation.

## Results

**Microstructural and near-atomic scale characterization.** Figure 1c–e presents a detailed APT analysis of the material aged

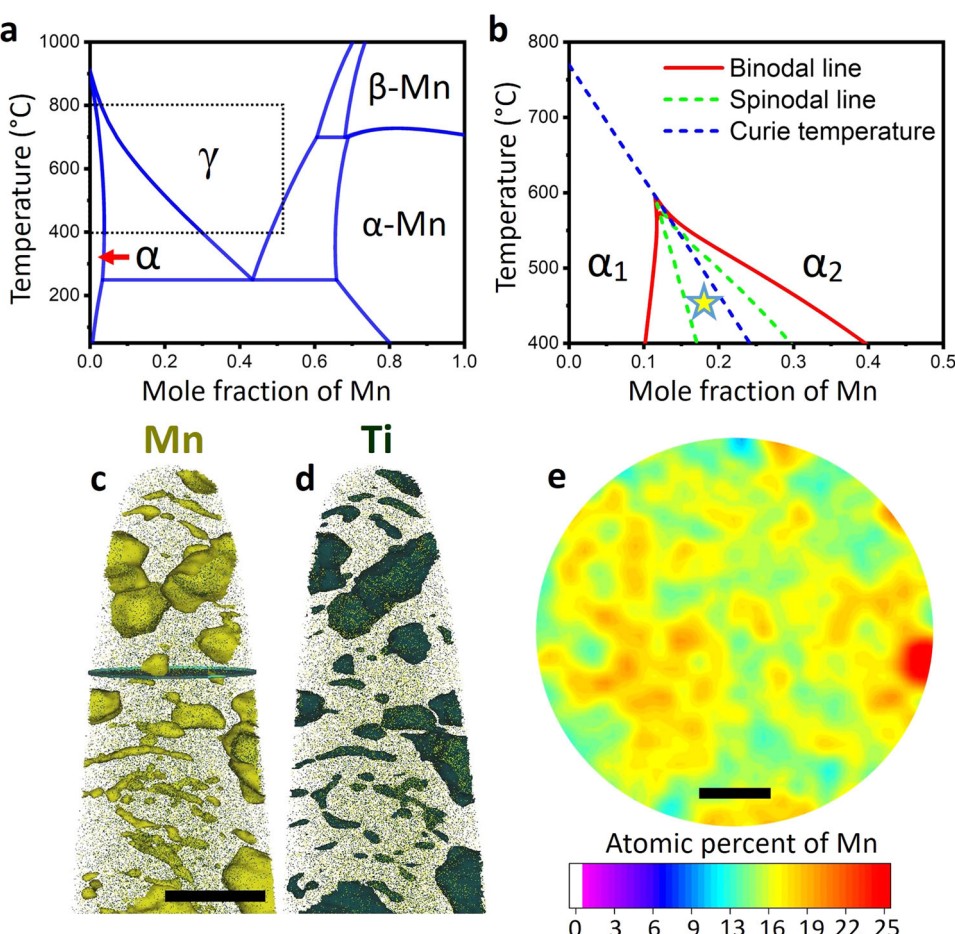

**Fig. 1 Materials design concept. a** Binary Fe-Mn phase diagram, including the α ferrite, γ austenite, α-Mn and β-Mn phases. **b** Metastable magnetic miscibility gap of the BCC phase. **c, d** APT reconstruction showing a 22 at% Mn iso-surface and a 5 at% Ti iso-surface revealing details of the precipitates and segregation to dislocations. Scale bar: 35 nm. **e** 2D compositional map (in at.%) of the cross section represented in **c** showing the extent of Mn compositional fluctuations in the matrix. Scale bar: 10 nm.

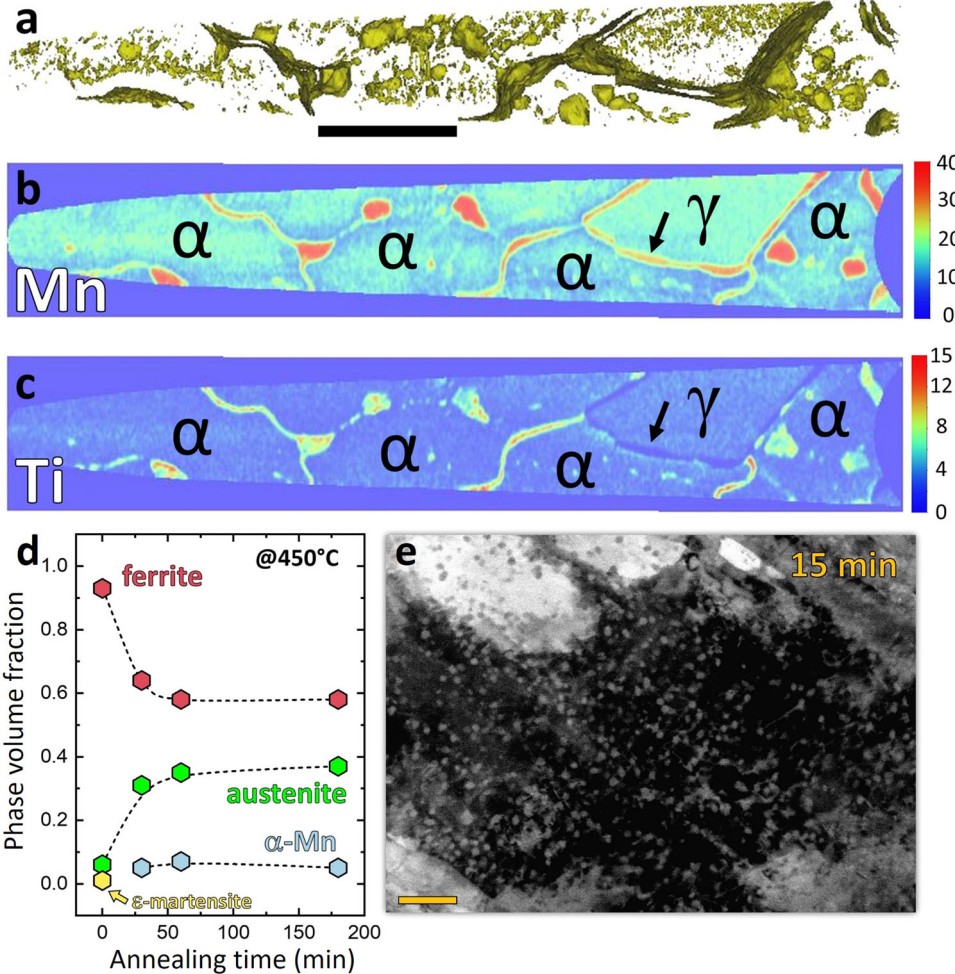

**Fig. 2 Microstructural analysis. a** APT reconstruction showing a 20at% Mn iso-surface. **b** 2D map of the Mn composition (in at.%). **c** 2D map of the Ti composition (in at.%). Scale bar, 100 nm. **d** Evolution of the fraction of the different phases after a given annealing time at 450 °C. **e** Microstructure after 15 min of aging at 450 °C. Scale bar, 200 nm.

during 15 min at 450 °C, evidencing the early-stages of phase decomposition. Figure 1c, d shows in detail a martensitic region where α-Mn precipitation is taking place, as revealed by the 22 at% Mn (1c) and 5 at%Ti (1d) iso-concentration surfaces. α-Mn precipitates nucleate at highly Mn-decorated dislocations, grain boundaries and in the bulk material. The identification of those elongated features revealed by the Mn and Ti iso-surfaces were the subject of extensive characterization in previous investigations for similar[12,13] and different[14] alloys. Figure 1e shows the extent of the Mn compositional fluctuations observed in the matrix. Following the initial concept of this alloy, the system is highly unstable against compositional fluctuations. These strong compositional fluctuations, observed in the bulk and at the dislocations and grain boundaries, act as precursors to the nucleation of the property-enhancing α-Mn precipitates.

Figure 2a shows the elemental distribution in a different region of the same microstructure as revealed by a set of iso-surfaces delineating regions containing 20 at% of Mn or more. Figure 2b, c presents 2D compositional maps of Mn and Ti in a cross section of the same specimen. The area identified as γ refers to an austenitic region which was originally retained after quenching and cold-rolling. It is possible to notice that the boundaries delimiting this region are depleted in Ti, unlike the α–α grain boundaries, dislocations and triple junctions. Upon heat treatment this austenite grows further, maintaining the local chemical equilibrium at the interface without requiring a new nucleation

event and incubation time[15]. Mn partitions to the growing austenite while Ti is depleted at the interface between austenite and ferrite, as indicated by the arrow, preventing the precipitation of α-Mn at this type of interface. This freshly formed reverted austenite is more stable than the original austenite, since it contains higher Mn (around 28 at%) and lower Ti (around 1.6 at%) content, contributing more effectively to increase the strain hardening of the material. Both Mn and Ti show strong tendency to segregate to α–α grain boundaries, triple junctions and dislocations enabling the nucleation of α-Mn at these defects. Figure 2d shows the fraction of different phases as a function of aging time, from X-ray diffraction. Prior to aging, the material was composed of α martensite plus a small fraction of retained austenite and hexagonal closed packed ε martensite. Upon aging, the growth of the retained austenite takes place in parallel to the precipitation of α-Mn (A12 structure), which is stabilized by the addition of Ti[11]. Scanning electron micrograph (electron channeling contrast imaging—ECCI) of the Fe18Mn3Ti alloy after 15 min@450 °C, in Fig. 2e, show homogenous precipitation of a second phase in the martensitic microstructure, identified as α-Mn phase A12.

High-resolution transmission electron microscopy (HRTEM) characterization was carried out for the Fe18Mn3Ti alloy aged during 6 h at 450 °C to allow coarsening for unequivocal characterization of the crystal structure and chemistry of the α-Mn precipitates. Figure 3a–d shows the chemical distribution

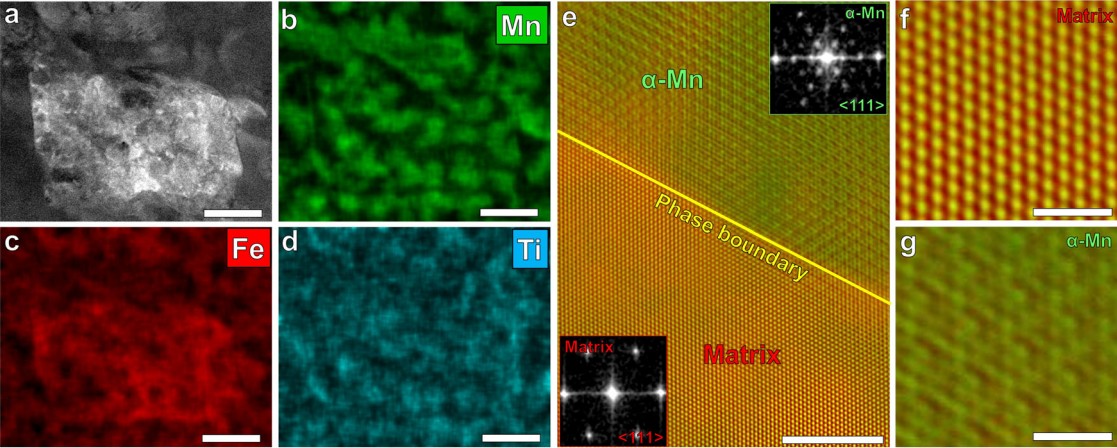

**Fig. 3 HRTEM characterization.** Fe18Mn3Ti alloy aged during 6 h@450 °C. **a–d** Energy-dispersive X-ray spectroscopy of a martensitic region with precipitation of α-Mn A12 phase (Mn-rich). Scale bar: 100 nm. **e** interface between BCC matrix (red) and α-Mn A12 phase (green). The insets are corresponding FFTs displays the cube-on-cube orientation relationship. Scale bar: 5 nm. **f, g** atomic resolution image of the BCC matrix and α-Mn A12 phase, respectively. Scale bar: 1 nm.

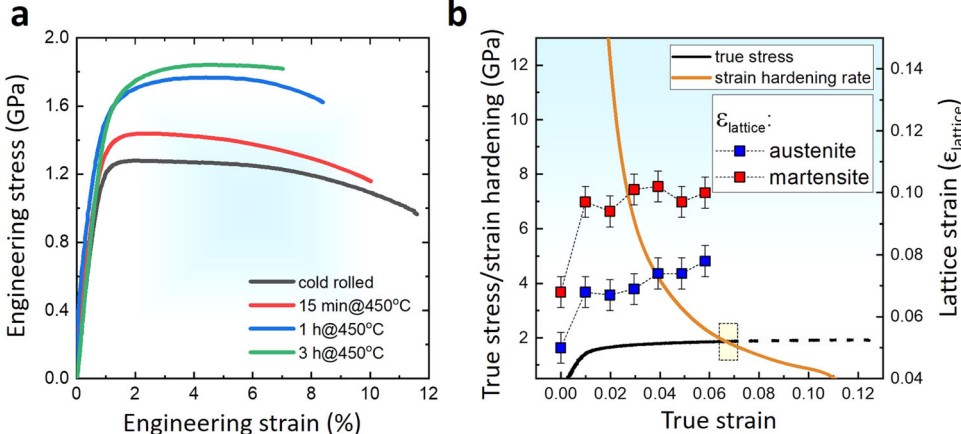

**Fig. 4 Mechanical performance and behavior. a** Tensile curves of the Fe18Mn3Ti alloy cold rolled and annealed up to 3 h at 450 °C. **b** in situ synchrotron XRD analysis of the Fe18Mn3Ti alloy cold rolled and annealed during 1 h at 450 °C showing the evolution of the lattice strain in the martensitic and austenitic phases during tensile deformation. Error bars represent the precision limit associated with measurements.

maps obtained by energy-dispersive X-ray spectroscopy (EDS) for Mn (b), Fe (c) and Ti (d) in the region in Fig. 3a showing the precipitation of α-Mn A12 phase (Mn-rich). Figure 3e shows a structural map obtained by high-resolution TEM of the microstructure after 6 h@450 °C. Fast Fourier transform (FFT) analysis shows an orientation relationship between BCC matrix and α-Mn phase A12 (cube-on-cube): $<111>_{BCCmatrix}//<111>_{A12}$; $\{110\bar{3}_{BCC\ matrix}//\{220\}_{A12}$. Figure 3f, g shows the atomic resolution image of the BCC matrix and α-Mn A12 phase, respectively. The α-Mn phase A12 has a unique complex body centered cubic (CBCC) structure which is based on the body-centered cubic unit cell, but contains 58 atoms representing four distinct positions[16].

**Mechanical properties.** Figure 4a shows engineering stress-strain curves of the Fe18Mn3Ti alloy aged up to 3 h@450 °C. Both, yield stress (YS) and ultimate tensile stress (UTS) systematically increase with increasing aging time indicating precipitation hardening of the martensitic matrix and higher work hardening due to the higher volume fraction of reverted austenite. We performed synchrotron XRD while in-situ tensile testing of the Fe18Mn3Ti alloy after 1 h@450 °C. This allows to evaluate in real-time the microstrain in austenite and martensite. The volume fraction of both phases was constant during the testing, indicating

the absence of a phase transformation induced plasticity effect (TRIP). Figure 4b shows the true stress vs true strain curve and the work-hardening rate as a function of the true strain. The lattice strain translates the local distortion imposed in the lattice planes by elastic and plastic deformation, enabling us to evaluate the strain partitioning among the phases. The lattice strain in martensite is higher than in austenite, since martensite inherited a highly deformed microstructure from cold-rolling and most of the austenite fraction was freshly reverted during heat treatment. The lattice strain in austenite remarkably increased from the yielding until the onset of necking at a true strain of 0.067 (or ε = 0.07) while the lattice strain remained nearly constant in martensite. Those results suggest that hardening of the martensite enables higher YS by constraining the deformation of austenite and a higher fraction of more stable austenite enables higher UTS through more extensive work hardening.

Since ultra-high strength materials with yield strength higher than 1500 MPa have typically low ductility, those materials must also be selected based on impact and fracture toughness. Maraging steels have superior toughness compared to conventional quenched and tempered ultra-high strength steels, such as AISI 4340 steel (possibly the most widely used ultra-high strength steel for mechanical construction), assuming similar yield

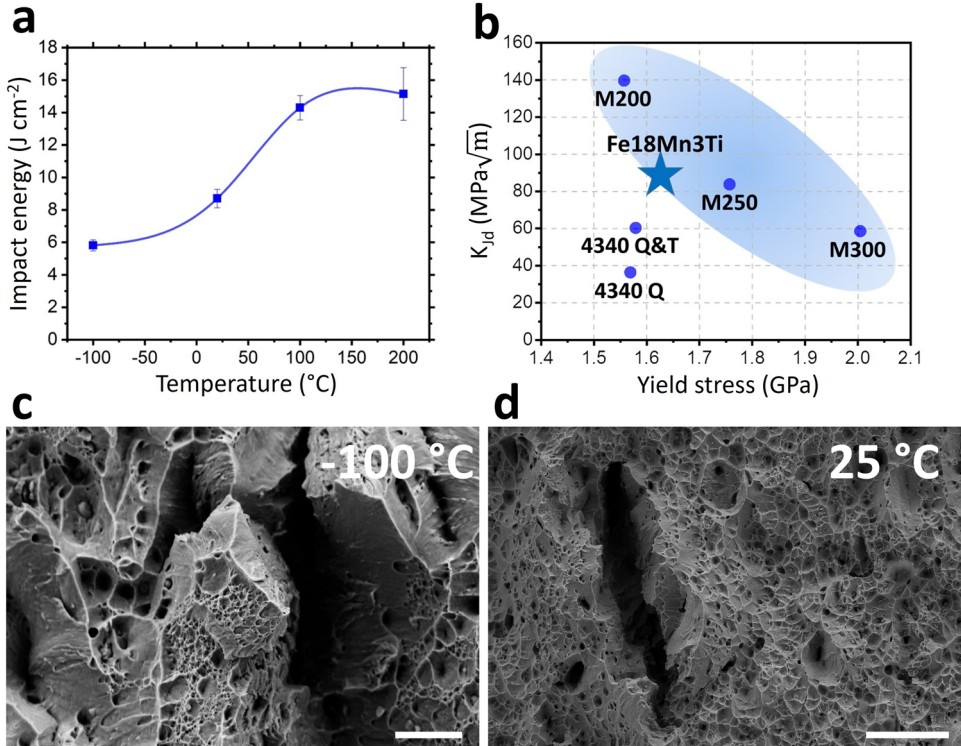

**Fig. 5 Impact toughness. a** Impact energy (subsize Charpy V-notch specimens) at different temperatures of the Fe18Mn3Ti alloy cold rolled and annealed during 1 h at 450 °C. Datapoints represent the average and standard deviation for 3 measurements. **b** *dynamic impact toughness* of the Fe18Mn3Ti alloy as compared to other ultra-high strenght steel grades. **c, d** Center regions of the fracture surfaces of the material after impact testing at −100 °C (**c**) and 25 °C (**d**). **c** Scale bar: 10 μm. **d** Scale bar: 40 μm.

strength values[17]. Therefore, we also performed impact testing at different temperatures using sub-sized Charpy V-notched (CVN) samples. Figure 5a shows the normalized impact energy of the Fe18Mn3Ti alloy aged during 1 h@450 °C (YS around 1600 MPa) for different testing temperatures. The material shows a gradual decrease in impact energy with decreasing temperature. This lack of an abrupt transition in impact energy absorption with decrease of temperature is also typical in FeNi-based maraging steels. Since the actual impact energy values are difficult to be compared to standard CVN specimens, we calculated the dynamic impact toughness ($K_{Jd}$) at room temperature. The Fe18Mn3Ti alloy aged during 1 h@450 °C displayed dynamic toughness ($88 \pm 12 \, MPa\sqrt{m}$) comparable to standard M250 maraging steel (with slightly higher yield stress), as displayed in Fig. 5b, assuming values reported in the literature[18]. Figure 5c, d shows the center region of the fracture surfaces of the material after impact testing at −100 °C and 25 °C, respectively. The presence of dimples across the whole surface of the material tested at 25 °C indicates that substantial plastic deformation took place before the fracture. The fracture surface of the material tested at −100 °C, on the other hand, displays characteristics of mixed ductile and brittle fracture, as evidenced by a reasonably smooth surface indicating cleavage coexisting with dimples indicating plasticity prior to fracture. Both fracture surfaces present larger voids possibly associated with the decohesion of austenite from the martensitic matrix.

## Discussion

Finally, although many other metallic alloys have been recently reported to exhibit excellent mechanical properties, their widespread and overall usefulness is often hindered by insufficient or even impossible upscaling of their production and manufacturing. These alloys may have an exceptional combination of

strength and toughness[19], nevertheless their application has been severely limited due to their prohibitive costs, caused mainly by their high Ni, Co and refractory elements contents. In Fig. 6a, b, we plot the estimated alloying cost and abundance risk level (ARL) of different ultra-high strenght steels and their respective tensile strength. The ARL was estimated based on the natural abundancy in Earth's crust (see Methods for definition). The alloys delimited by the blue line are high-Co maraging steels. Two high-Co (FeCoCrNiMn and CoNiCr) multicomponent materials, referred to as high- or medium-entropy alloys have also been added. The alloying costs for these materials are estimated to be up to three times higher than the alloying costs for the most expensive Co-containing maraging steels without delivering comparable properties. Our Fe18Mn3Ti alloy was found to have similar tensile strength as compared to the low Co grades of Ni-containing maraging steels which contain up to 4 wt.% of Mo for intermetallic precipitation. These diagrams clearly show the importance of lean-alloy design concepts, such as applied here for our new Fe18Mn3Ti alloy. The comparison reveals that highest benefit for a more sustainable society can be reached when reconciling key engineering material features such as weight reduction in transportation, to reduce energy consuption by using ultra-high strenght materials, with a more responsible use of alloying elements that enable such materials.

## Methods

**Alloy design and production.** The thermodynamic quantities and phase diagram were calculated using the Thermocalc software together with the TCFE9 thermodynamic database for iron solid solutions. Details of the models used in this database can be found in refs. [20–22]. The central design concept of this paper is to select an alloy composition which would be unstable against composition fluctuations to allow homogeneous nucleation of precipitates. Supplementary Fig. 1a shows the stability function for our alloy composition in function of the temperature which was selected to have a minimum value at our intended annealing

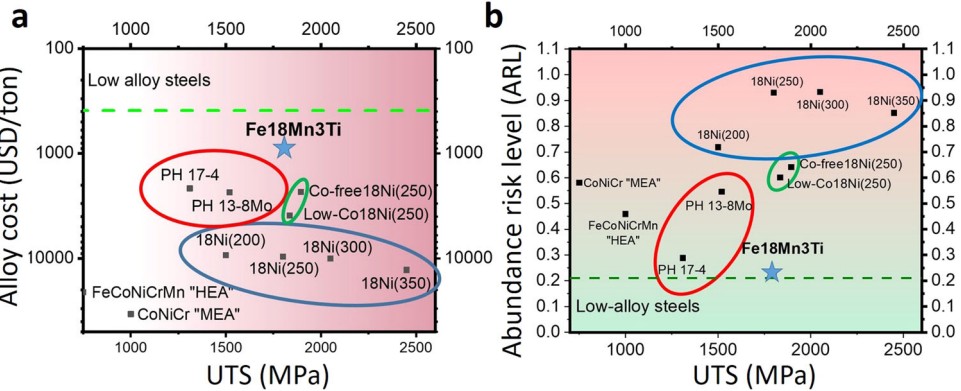

**Fig. 6 Potential for application. a** Alloy cost comparison and (**b**) abundance risk level (ARL) of different precipitation hardened ultra-high strenght steels.

temperature. The Hessian matrix[23,24], which describes the local curvature of the Gibbs energy curve, is obtained by calculating the second derivatives $\Omega_{ij}$ of the Gibbs energy G of a given phase with respect to mole fractions ($x_i$) of the components:

$$\Omega_{ij} = \left( \frac{\partial^2 G}{\partial x_i \partial x_j} \right)_{T,p,x_k} \tag{1}$$

A phase is stable against compositional fluctuations if det$|\Omega| > 0$ and the locus where the stability function changes sign (det$|\Omega| = 0$) is called the spinodal. A solution can endure spinodal decomposition when it is unstable against compositional fluctuations, i.e. when the stability function is negative (det$|\Omega| < 0$). In the Thermocalc software, the stability function (QF) is the lowest of the Eigenvalues ($e_n$) obtained by the diagonalization of the Hessian matrix divided by the lowest eigenvalue ($se_n$) obtainned for a corresponding ideal phase:

$$QF(ph) = \min(\{e_1, e_2, \dots, e_n\})/\min(\{se_1, se_2, \dots, se_n\}) \tag{2}$$

A second challenge for our alloy design strategy was to obtain an alloy composition and processing route which would lead to a very high fraction of BCC alpha martensite which is the phase where the metastable phase separation between α1 and α2 is possible. Typically, compositions with very high Mn content lead to microstructures containing a high fraction of retained austenite and epsilon martensite since Mn is a strong austenite stabilizing element. We added 3wt.% Ti to maximize the amount of metastable alpha martensite in the microstructure following the strategy adopted by ref. [10]. Supplementary Fig. 1b shows the equilibrium pseudo-binary phase diagram for the Fe–18Mn–$x$Ti (wt.%) composition calculated using the HMnS03 database[25]. The addition of only 3wt.% Ti secures the existence of a single-phase gamma field wide enough to allow homogenization and a solid solution state without precipitation of Laves (C14) phase. We should also mention that there is a lack of thermodynamic assessments for the low temperature equilibria of the Fe–Mn–Ti system. Notwithstanding, the existing assessments from high temperature equilibria indicate that Ti is an α-Mn stabilizing element, but also a much stronger stabilizer of the Laves (C14) phase. Indeed, both databases, TCFE9 and HMnS03, predict Laves as an equilibrium phase at 450 °C, coexisting with ferrite (iron-rich BCC) and austenite. Therefore, α-Mn is likely a metastable phase formed by the combination of our multi-step nucleation pathway and the lower energy of the coherent interface between α-Mn and the surrounding BCC matrix (as compared with the incoherent interface between Laves and the BCC matrix). The material was annealed up to 2 weeks at 450 °C and we did not observe a volume fraction of Laves phase significant enough to be detected by XRD.

An ingot of the Fe–18%Mn–3%Ti (wt.%) alloy used in this work was first synthesized in a vacuum induction furnace and cast as a 60 cm×40 cm rectangular billet using high-purity metals to obtain a high standard of cleanliness comparable to Nickel maraging steels. The highly segregated edges of the slab were subsequently cut off and the material was hot-rolled from 60 to 6 mm at 1100 °C. The material was solution treated at 1100 °C during 3 h and then water quenched. The material was subsequently quenched in liquid nitrogen during 30 min and then cold-rolled to 55% thickness reduction for reducing retained austenite fraction and increasing the dislocation density. The material was finally aged at 450 °C for 15 min, 30 min, 60 min and 180 min. The nominal chemical composition of the alloy was measured by wet-chemical analysis (Supplementary Table 1).

**Analytical methods**. Scanning electron microscopy (SEM) was performed in a Zeiss Merlin (Carl Zeiss AG) featuring a Gemini 2-type field emission gun (FEG) electron column. The samples were further characterized by high-resolution X-ray diffraction (XRD) using a Seifert Type ID3003 Diffractometer and Co-Kα1 radiation (λ = 1.78897 Å) using a 1.5 mm beam size. The scanning range, rate and step size were 20–130°, 20 s/step and 0.03°, respectively. The volume fraction of

ferrite, austenite and α-Mn was calculated using the Rietveld simulation method performed with the software MAUD (Materials Analysis Using Diffraction).

APT specimens were prepared using an FEI Helios NanoLab600i dual-beam Focused Ion Beam (FIB)/Scanning Electron Microscopy (SEM) instrument. APT was performed using a LEAP 500XS device by Cameca Scientific Instruments, with approximately 80% detection rate efficiency, at a set-point temperature of 60 K in laser-pulsing mode at a wavelength of 355 nm, 500 KHz pulse repetition rate, and 40 pJ pulse energy. For reconstructing 3D atom maps, visualization, and quantification of segregation, the commercial software IVAS® by Cameca was employed following the protocol introduced by Geiser et al.[26] and detailed in Gault et al.[27] Voltage mode was used to reconstruct the datasets.

The transmission electron microscope (TEM) foils were prepared by a dual-beam focus ion beam (FIB FEI 600i instrument) via TEM lift-out procedures. High resolution TEM (HRTEM), high angle annular dark field (HAADF) scanning TEM (STEM) and energy dispersive X-ray spectroscopy (EDS) were carried out employing a using an image aberration corrected TEM (FEI Titan Themis). For HAADF-STEM imaging, a probe semiconvergence angle of 17 mrad and inner and outer semi-collection angles ranging from 73 to 200 mrad were operated.

**Mechanical testing**. Tensile testing was performed using a Kammrath and Weiss stage. The strain was measured by digital image correlation (DIC) using the Aramis software (GOM GmbH). Three flat samples were analyzed in total from each aging condition at room temperature and an initial strain rate of $10^{-3}$ s$^{-1}$. The thickness, width and gauge length of the samples were 1 mm, 2 mm and 8 mm, respectively. Vickers hardness (HV5) measurements shown in Supplementary Fig. 5 were performed using a LECO M-400-G (LECO Instrumente).

**Mechanical testing during in situ synchrotron XRD**. Tensile testing was also performed during in-situ synchrotron XRD measurements at the Brazilian Synchrotron Light Laboratory (LNLS), using the XTMS workstation[28]. We used a thermomechanical simulator Gleeble™ 3S50 coupled to a monochromatic synchrotron beam whose wavelength (λ) and energy are 0.103323 nm and 12 keV, respectively. Tensile specimens were positioned inside the thermomechanical simulator with an inclination of 15° related to the incident beam. Data acquisition were conducted using a 2D solid-state Rayonix® (model SX165) detector, placed at a fixed distance of 285 mm from the specimen and displaying a 2θ angle of 40°. Debye-Scherrer cones were collected within the η interval between −25° and 25°. Extended Data Fig. 3 shows the schematic representation of the setup geometry as well as the dimensions of the tensile specimens. Further details are outlined in[28]. Before testing, the setup was aligned, and the instrumental contribution was determined using standard $Y_2O_3$ powders.

The material was deformed under a constant crosshead displacement speed of $1.7 \times 10^{-6}$ m s$^{-1}$ up to fracture at an engineering strain of ε = 0.13. Debye-Scherrer cones were collected at a rate of 3 images per minute. Plots of intensity versus the diffraction angle 2θ were constructed by integrating the 2D diffraction data with the aid of the software DRACON (Diffracted X-Ray Analysis Console)[28]. The software Dracon (Diffracted X-Ray Analysis Console) was used to integrate the 2D diffraction data and convert them into plots of intensity versus the diffraction 2θ angle. The obtained diffractograms were subjected to fitting profile analysis using a pseudo-Voigt function to obtain information regarding peak position, integrated intensity, and the values of full width at half maximum (FWHM). The Modified Williamson-Hall method[29] was employed to assess the magnitude of the lattice strain (ε lattice) using

$$\triangle K \cong \frac{1}{D} + \varepsilon_{lattice} \cdot \left[ K^2 \cdot \bar{C}_{h00} \cdot \left(1 - qH^2\right) \right] \tag{3}$$

where $\triangle K = 2\cos\theta(FWHM)/\lambda$, D is the size of the crystallite portions responsible for diffracting the radiation, $K = 2\sin\theta/\lambda$, ε lattice is the lattice strain and $\bar{C}_{h00}$ is a constant related to the elastic constants of the material[29,30]. $H^2$ is given by

$(h^2k^2 + h^2l^2 + k^2l^2)/(h + k + l)^2$ and $q$ indicates the character of the dislocations[29,30]. Here, $\bar{C}_{h00}$ was assumed as 0.332 and 0.285, respectively, for austenite and α'-martensite as in preceding studies[28]. The magnitude of the lattice strain ($\varepsilon_{lattice}$) was determined by ensuring the linearity between $\triangle K$ and $K^2 \cdot \bar{C}_{h00} \cdot (1 - qH^2)$[21]. The values of $q$ were taken as those which provided the best linear fit for Eq. (1). Thus, the values of $\varepsilon_{lattice}$ were taken from the slope of the modified Williamson–Hall plots (Supplementary Fig. 4).

**Impact testing**. Instrumented impact testing was performed using a Zwick/Roell RKP450 machine using sub-sized Charpy V-notched (CVN) samples (Supplementary Fig. 5a). The J-integral (strain energy release rate) was calculated using ASTM E1820-17a standard. The integral is computed as the sum of elastic, $J_{el(i)}$, and plastic components, $J_{pl(i)}$, for each data point $i$:

$$J_{(i)} = J_{el(i)} + J_{pl(i)} = \frac{K_{(i)}^2(1 - \nu^2)}{E} + J_{pl(i)} \quad (4)$$

where $E$ is Young's modulus, ν is Poisson's ratio, and $K_{(i)}$ is the stress intensity factor corresponding to each data point, which is given by

$$K_{(i)} = \left[\frac{P_i S}{(BB_N)^{\frac{1}{2}}W^{\frac{3}{2}}}\right]f\left(\frac{a_i}{W}\right) \quad (5)$$

where $P_i$ is the applied load at each individual data point, $S$ is the spacing between the supports of the impact machine, $B$ is the starting sample thickness, $B_N$ is the sample thickness at each individual data point, W is the sample width, and $f\left(\frac{a_i}{W}\right)$ is a geometry-dependent function of the ratio of crack length, $a_i$, to width, W, as listed in the ASTM standard. The plastic component $J_{pl(i)}$ is given by the following equation:

$$J_{pl(i)} = \left[J_{pl(i-1)} + \left(\frac{\eta_{pl(i-1)}}{b_{i-1}}\right)\left(\frac{A_{pl(i)} - A_{pl(i-1)}}{B_N}\right)\right]\left[1 - \gamma_{pl(i-1)}\left(\frac{a_i - a_{i-1}}{B_N}\right)\right] \quad (6)$$

where $\eta_{pl(i-1)} = 2 + 0.522\, b_{(i-1)}/W$ and $\gamma_{pl(i-1)} = 1 + 0.76\, b_{(i-1)}/W$. $A_{pl(i)} - A_{pl(i-1)}$ is the increment of plastic area underneath the load–displacement curve, and $b_i$ is the uncracked ligament width (i.e., $b_i = W - a_i$). The dynamic fracture toughness expressed in terms of the stress intensity was then computed using the standard J–K equivalence (mode I) relationship $K_{Jd} = (E'\, J_d)^{1/2}$, assuming $E' = E/(1-\nu^2)$, at the maximum force $P_{max}$ during the impact testing (Supplementary Fig. 5b).

**Alloy cost**. The alloy cost was calculated using the estimated cost by alloy addition (USD/wt.% per tonne) found in ref. [31]. Those values fluctuate considerably according to demand[32] and they are used here only for the sake of relative comparison. Supplementary Fig. 6 shows the historical price of cobalt, nickel, titanium and ferromanganese according to references[32–34]. Supplementary Table 2 shows the alloy composition of different steels according to reference[17]. Tensile strength data was obtained from the same reference for the different steels and from reference[35] for the FeCoNiCrMn and CoNiCr alloys.

**Abundance risk level (ARL)**. We define the abundance risk level (ARL) of a given element as the inverse of the LOG of its abundance in the earth's crust [ppm]:

$$ARL_{element} = 1/LOG[abundance] \quad (7)$$

The Abundance Risk Level of a material is given by the summation of the products of the weight fraction $w_i$ of the different alloying elements by the ARL of the respective elements:

$$ARL_{alloy} = \sum_i w_i ARL_i \quad (8)$$

## Data availability

All relevant data supporting the findings of this study are contained in the paper and its Supplementary Information files. All other relevant data are available from the corresponding author (A.K.d.S.) upon reasonable request.

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

## Acknowledgements

The authors are grateful to U. Tezins and A. Sturm for their support to the FIB and APT facilities at MPIE, to B. Breitbach for his support with XRD characterization and to M. Adamek for his help with tensile tests. Authors are grateful to LabNano of the National Research Center for Energy and Materials (CNPEM – Campinas-SP, Brazil) for allowing the use of their facilities, and to Mr. L. Wu for his technical support. A.K.d.S. is grateful to the Brazilian National Research Council (Conselho Nacional de Pesquisas, CNPQ) for financial support through the "Science without Borders" Project (203077/2014-8). IRSF acknowledges financial support through Capes-Humboldt (grant number 88881.512949/2020-01).

## Author contributions

A.K.d.S. was the lead scientist of the study and proposed the alloy design concept; A.K.d.S. and IRSF performed the SEM and XRD characterization; A.K.d.S. and B.G. performed the APT experiments and analysis; I.R.S.F. and K.Z. performed the synchrotron experiments and analysis; W.L. performed the TEM experiments; A.K.d.S., M.F.H., L.M.A., D.P., and I.R.S.F. analyzed the mechanical behavior of the material; A.K.d.S., D.R., and B.G. wrote the paper. All authors contributed to reviewing and editing the manuscript and discussing and interpreting all the results.

## Funding

## Competing interests

The authors declare no competing interests.
