## [Peer Review File · Nature Communications]

Title: A sustainable ultra-high strength Fe₁₈Mn₃Ti maraging steel through controlled solute segregation and α -Mn nanoprecipitationREVIEWER COMMENTS

Reviewer #1 (Remarks to the Author):

This is an interesting work for developing a new type of high ultra-high strength steel with very simple composition and much low cost compared with the current maraging steels. The principle for strengthening of the steel is related a novel mechanism, segregation of the composition (Mn in this study), which has been seldom reported. This type new steel will have great application values if its other properties are comparable with the current ultra-high strength steels. However, there are somethings need to be cleared before the consideration of publication:

1. How the composition of Fe₁₈Mn₃Ti was selected as the experimental material is not clear.
2. What the role of Ti addition is needs to be further stated.
3. The microstructure of Fe₁₈Mn₃Ti shown in Fig.2 (e) is too blurred to see clearly, even without the scale bar.
4. It will be better to conduct a high resolution TEM observation on the Mg segregation and others in Fe₁₈Mn₃Ti for further confirmation of the strengthening mechanism.

Reviewer #2 (Remarks to the Author):

Dear authors,

Authors investigated the interesting microstructural evolution and the excellent mechanical property of Fe–18Mn–3Ti maraging steel. This new type advanced high-strength steel is well designed base on the author's strategy, and its microstructure and mechanical properties were characterized carefully by using some advanced techniques. However, this article seems to be a chip show of author's research. For instance, although, spinodal decomposition between alpha-1 and -2 in Fe–Mn system leading to very fine precipitation is an originality of this article, it had been already reported by authors (2018, Nature Comm.). Therefore, the reviewer judged it is not appropriate as regular article published in Nature Communications.

Sincerely,

Reviewer #3 (Remarks to the Author):

Comments to the Author:

This study developed an ultra-high strength Fe₁₈Mn₃Ti maraging steel through controlled solute segregation and α -Mn nanoprecipitation. The idea is quite new and amazing, which can strengthen the steel greatly without adding critical elements. However, there are several points should be addressed

clearly before the next submission.

1. In Fig.1, the authors stated that α -Mn is observed from the APT results. However, it seems that Mn and Ti are co-segregated. How to remove the possibility of Mn-Ti intermetallic compound formation? On the other hand, how to determine the precipitates and segregation to dislocations solely from the APT results in Fig. 1?
2. "The area identified as γ refers to an austenitic region which was originally retained after quenching and cold-rolling." How to identify the gamma phase from APT results in Fig. 2? It seems that it has no big difference in Mn or Ti concentration with the matrix. Or the authors indicate the gamma phase by the arrow? Please make it clearly. If the retained austenite is formed during quenching, the Ti content may be the same as the matrix.
3. In lines 83-85, "Upon heat treatment this austenite grows further keeping the local chemical equilibrium at the interface. Mn partitions to the growing austenite while Ti is depleted at the interface between austenite and ferrite". Do the authors have any evidence to say so ?
4. The authors showed the phase fraction in Fig. 2d measured by XRD. However, the α -Mn nanoprecipitation is quite fine and as low as around 5% according to the authors results. How the authors could be confirmed that the nano particles could be measured accurately by XRD.
5. "Scanning electron micrograph of the Fe₁₈Mn₃Ti alloy after 60min@450 °C, in Figures 2e, show homogenous precipitation of a second phase in the martensitic microstructure, identified as α -Mn phase A12." It is quite hard to identify the precipitation in TEM image in Fig. 2(e), which is quite important for the readers to get the overall image of α -Mn nano particles. Please provide clearer and typical image.

We greatly appreciate the suggestions of the reviewers and we thoroughly revised our manuscript (NCOMMS-21-37589), entitled

“Ultra-high strength Fe₁₈Mn₃Ti maraging steel through controlled solute segregation and α -Mn nanoprecipitation”

submitted before to *Nature Communications* based on reviewer’s comments. The new title of the manuscript is:

“A sustainable ultra-high strength Fe₁₈Mn₃Ti maraging steel through controlled solute segregation and α -Mn nanoprecipitation”.

The answers to the reviewer’s comments are as follows.

Reviewer #1

Opening comment: “This is an interesting work for developing a new type of high ultra-high strength steel with very simple composition and much low cost compared with the current maraging steels. The principle for strengthening of the steel is related a novel mechanism, segregation of the composition (Mn in this study), which has been seldom reported. This type new steel will have great application values if its other properties are comparable with the current ultra-high strength steels. However, there are somethings need to be cleared before the consideration of publication: (...)”

Answer: We would like to cordially thank the reviewer for the strong support and important comments.

Comment 1: “How the composition of Fe₁₈Mn₃Ti was selected as the experimental material is not clear.”

Answer: Thank you for this comment. We selected a composition in the middle of the metastable miscibility gap to probe the hypothesis that Mn segregation act as a precursor for second phase nucleation. Our previous works stressed that Mn segregation to grain boundaries and dislocations provide a non-classical nucleation pathway for the austenite. If this hypothesis is correct, it should also follow that having segregation taking place in the whole bulk should lead to a homogenous precipitation. We just followed this hypothesis, but we had to find a way to obtain a BCC phase supersaturated with Mn but with low fraction of retained austenite. We achieved that by adding 3wt% Ti and by cold rolling of the material.

Comment 2: “What the role of Ti addition is needs to be further stated.”

Answer: Thank you for the pertinent suggestion. We added 3wt.% Ti mostly to avoid a high fraction of retained austenite and epsilon martensite after quenching and cold-rolling. This strategy was

previously adopted by *Paduani et al.* Journal of Applied Physics 70, 7524 (1991) in order to stabilize and retain the α -phase at room temperature to determine the dependence of the average hyperfine field and critical temperature (T_c) with Mn content. We added the following description about the role of Ti addition to the Methods session:

“A second challenge for our alloy design strategy was to obtain an alloy composition and processing route which would lead to a very high fraction of BCC alpha martensite which is the phase where the metastable phase separation between α_1 and α_2 is possible. Typically, compositions with very high Mn content lead to microstructures containing a high fraction of retained austenite and epsilon martensite since Mn is a strong austenite stabilizing element. We added 3wt.% Ti to maximize the amount of metastable alpha martensite in the microstructure following the strategy adopted by ref.16. Supplementary Fig. 1b shows the equilibrium pseudo-binary phase diagram for the Fe-18Mn-xTi (wt.%) composition calculated using the HMnS03 database³⁰. The addition of only 3wt.% Ti secures the existence of a single-phase gamma field wide enough to allow homogenization and a solid solution state without precipitation of Laves (C14) phase. We should also mention that there is a lack of thermodynamic assessments for the low temperature equilibria of the Fe-Mn-Ti system. Notwithstanding, the existing assessments from high temperature equilibria indicate that Ti is a α -Mn stabilizing element, but also a much stronger stabilizer of the Laves (C14) phase. Indeed, both data bases, TCFE9 and HMnS03, predict Laves as an equilibrium phase at 450°C, coexisting with ferrite (iron-rich BCC) and austenite. Therefore, α -Mn is likely a metastable phase formed by the combination of our multi-step nucleation pathway and the lower energy of the coherent interface between α -Mn and the surrounding BCC matrix (as compared with the incoherent interface between Laves and the BCC matrix). The material was annealed up to 2 weeks at 450°C and we did not observe a volume fraction of Laves phase significant enough to be detected by XRD.”

Comment 3: “The microstructure of Fe18Mn3Ti shown in Fig.2 (e) is too blurred to see clearly, even without the scale bar.”

Answer: Thanks for pointing out this problem. We believe the Fig.2(e) was blurred due to the conversion to the pdf format. We updated Fig.2(e) and we added additional HRTEM results.

Comment 4: “It will be better to conduct a high resolution TEM observation on the Mg segregation and others in Fe18Mn3Ti for further confirmation of the strengthening mechanism.”

Answer: We thank the reviewer for the suggestion. We fully agree and we already had additional HRTEM characterization conducted which we now have added to the results as the new Figure 3: HRTEM characterization.

Reviewer #2

Opening comment: “Authors investigated the interesting microstructural evolution and the excellent mechanical property of Fe–18Mn–3Ti maraging steel. This new type advanced high-strength steel is well designed base on the author’s strategy, and its microstructure and mechanical properties were characterized carefully by using some advanced techniques.”

Answer: We thank the reviewer for appreciating our work.

Comment 1: “However, this article seems to be a chip show of author’s research. For instance, although, spinodal decomposition between alpha-1 and -2 in Fe–Mn system leading to very fine precipitation is an originality of this article, it had been already reported by authors (2018, Nature Comm.). Therefore, the reviewer judged it is not appropriate as regular article published in Nature Communications.”

Answer: We thank the reviewer for this comment. Yet, the reviewer might have slightly misinterpreted our previous report in Nat. comm. 9 (1), 1-11, where we reported confined spinodal fluctuations at grain boundaries and dislocations in FeMn alloys. We do not report any “spinodal decomposition between alpha-1 and -2 in Fe–Mn system leading to very fine precipitation” in this previous publication, but we reported confined spinodal fluctuations at grain boundaries and dislocations in FeMn alloys. The proposed mechanism consisted “in solute adsorption (or segregation) to crystalline defects followed by linear and planar spinodal fluctuations in an Fe-Mn model alloy. These fluctuations provide a pathway for austenite nucleation due to the higher driving force for phase transition in the solute-rich regions”. The theory of this mechanism was further developed in Acta Materialia 168, 109-120 and npj Computational Materials 6, 191. It is about the thermodynamics at confined lattice defect spaces and has nothing to do with maraging steels.

In the current work, we did not target the chemical decoration of grain boundaries and dislocations, but we target precipitation in a bulk system. More specific, we shifted the alloy’s bulk composition beyond the thermodynamic stability limit should lead to a fine precipitation which would lead to the

strengthening of the material. We selected a lean composition and processing route which would allow a high fraction of martensite supersaturated with Mn and we tested those hypotheses by extensive microstructural and mechanical characterization. Therefore, we disagree with the reviewer's comment concerning the originality of our work.

Reviewer #3

Opening comment: “This study developed an ultra-high strength Fe18Mn3Ti maraging steel through controlled solute segregation and α -Mn nanoprecipitation. The idea is quite new and amazing, which can strengthen the steel greatly without adding critical elements. However, there are several points should be addressed clearly before the next submission.”

Answer: We would like to cordially thank the reviewer for the strong support and important comments.

Comment 1: “In Fig.1, the authors stated that α -Mn is observed from the APT results. However, it seems that Mn and Ti are co-segregated. How to remove the possibility of Mn-Ti intermetallic compound formation? On the other hand, how to determine the precipitates and segregation to dislocations solely from the APT results in Fig. 1?”

Answer: We thank the reviewer for this hint. Yes, indeed Mn and Ti are co-segregating and that is possibly the reason why the segregation of Mn and Ti leads to the nucleation of α -Mn instead of austenite, since Ti is a α -Mn stabilizer element. We also added the following text to the Methods session concerning the possibility of intermetallic (Laves phase) formation:

"We added 3wt.% Ti to maximize the amount of metastable alpha martensite in the microstructure following the strategy adopted by ref.16. Supplementary Fig. 1b shows the equilibrium pseudo-binary phase diagram for the Fe-18Mn-xTi (wt.%) composition calculated using the HMnS03 database³⁰. The addition of only 3wt.% Ti secures the existence of a single-phase gamma field wide enough to allow homogenization and a solid solution state without precipitation of Laves (C14) phase. We should also mention that there is a lack of thermodynamic assessments for the low temperature equilibria of the Fe-Mn-Ti system. Notwithstanding, the existing assessments from high temperature equilibria indicate that Ti is a α -Mn stabilizing element, but also a much stronger stabilizer of the Laves (C14) phase. Indeed, both data bases, TCFE9 and HMnS03, predict Laves as an equilibrium phase at 450°C, coexisting with ferrite (iron-rich BCC) and austenite. Therefore, α -Mn is likely a

metastable phase formed by the combination of our multi-step nucleation pathway and the lower energy of the coherent interface between α -Mn and the surrounding BCC matrix (as compared with the incoherent interface between Laves and the BCC matrix). The material was annealed up to 2 weeks at 450°C and we did not observe a volume fraction of Laves phase significant enough to be detected by XRD.”

Comment 1.1: On the other hand, how to determine the precipitates and segregation to dislocations solely from the APT results in Fig. 1?”

Answer: The identification of those elongated features with dislocations enriched with solute was the focus of previous publications. We added those publications as a reference in the text. Precipitates can be distinguished from Mn-enriched dislocations based on their size, shape and composition. Concerning their size and shape, they are not confined to a small region resembling the elongated features or grain boundaries, but they can further grow and coarsen over time. Concerning their composition, they contain higher amounts of Mn and Ti as compared to the decorated dislocations and compositional fluctuations at the matrix (please, note the relatively large α -Mn precipitates at triple junctions in Figure 2 and how they differ in size, shape and composition from the adjoining grain boundary).

Comment 2: “The area identified as γ refers to an austenitic region which was originally retained after quenching and cold-rolling.” How to identify the gamma phase from APT results in Fig. 2? It seems that it has no big difference in Mn or Ti concentration with the matrix. Or the authors indicate the gamma phase by the arrow? Please make it clearly. If the retained austenite is formed during quenching, the Ti content may be the same as the matrix.”

Answer: We improved the description in the following way: “The area identified as γ refers to an austenitic region which was originally retained after quenching and cold-rolling. It is possible to notice that the boundaries delimiting this region are depleted in Ti, unlike the α - α grain boundaries, dislocations and triple junctions. Upon heat treatment this austenite grows further, maintaining the local chemical equilibrium at the interface without requiring a new nucleation event and incubation time. Mn partitions to the growing austenite while Ti is depleted at the interface between austenite and ferrite, as indicated by the arrow, preventing the precipitation of α -Mn at this type of interface. This freshly formed reverted austenite is more stable than the original austenite, since it contains a

higher Mn (around 28 at%) and lower Ti (around 1.6 at%) content, contributing more effectively to increase the strain hardening of the material.”

Comment 3: “In lines 83-85, “Upon heat treatment this austenite grows further keeping the local chemical equilibrium at the interface. Mn partitions to the growing austenite while Ti is depleted at the interface between austenite and ferrite”. Do the authors have any evidence to say so?”

Answer: We thank the reviewer for this comment. We can observe the partitioning of the elements at the moving gamma interface in Figure 2 which is still an early stage of annealing (15 min). We also measure the austenite fraction as a function of the aging time (Figure 2d) by X-ray diffraction. Those observations agree with the expected behavior about the growth of the retained austenite during intercritical annealing as reported in previous publications (e.g.: O. Dmitrieva *et al.* *Chemical gradients across phase boundaries between martensite and austenite in steel studied by atom probe tomography and simulation. Acta Materialia* 59, 1, 2011)

Comment 4: “The authors showed the phase fraction in Fig. 2d measured by XRD. However, the α -Mn nanoprecipitation is quite fine and as low as around 5% according to the authors results. How the authors could be confirmed that the nano particles could be measured accurately by XRD.”

Answer: We would like to thank the reviewer for this comment. The volume fraction of ferrite, austenite and α -Mn was calculated using the Rietveld simulation method performed with the software MAUD (Materials Analysis Using Diffraction). We obtained a GoF (goodness of fitting) of 1.1 (ideally 1) between our simulated and experimental diffractograms (“Figure a” below for the Fe₁₈Mn₃Ti alloy aged during 3h@450°C). It was also possible to do the deconvolution of the peaks related to the 3 different phases (“Figure b” below for the Fe₁₈Mn₃Ti alloy aged during 3h@450°C).

Comment 5: “Scanning electron micrograph of the Fe₁₈Mn₃Ti alloy after 60min@450 °C, in Figures 2e, show homogenous precipitation of a second phase in the martensitic microstructure, identified as α -Mn phase A12.” It is quite hard to identify the precipitation in TEM image in Fig. 2(e), which is quite important for the readers to get the overall image of α -Mn nano particles. Please provide clearer and typical image.”

Answer: We would like to thank the reviewer for this comment. The previous Figure 2e was blurred due to the file conversion to pdf. We added a higher quality image to Figure 2e and we added more detailed HRTEM characterization images to Figure 3.

REVIEWERS' COMMENTS

Reviewer #1 (Remarks to the Author):

This work provided a new strategy to fabricate maraging steel with much cheaper strengthening element, only Mn and Ti, without Ni and Co that are expensive but conventionally used in maraging steels. The chemical composition of the steel was smartly designed through thermodynamic calculation and the microstructure of martensitic matrix with nano-sized Mn-rich precipitations was verified by APT and HRTEM analyses. The strength has reached the level of conventional maraging steels, though that of some maraging steels is over 2GPa. Thus the new steel with low cost has much potential for applications. One question is how the impurities in the steel were controlled, as those of the conventional maraging steels?

Dear Editor,

We greatly appreciate the suggestions of the reviewer and we revised our manuscript (NCOMMS-21-37589A), entitled

“A sustainable ultra-high strength Fe₁₈Mn₃Ti maraging steel through controlled solute segregation and α -Mn nanoprecipitation”.

The answers to the reviewer’s comments are as follows.

Reviewer #1

Comment: “This work provided a new strategy to fabricate maraging steel with much cheaper strengthening element, only Mn and Ti, without Ni and Co that are expensive but conventionally used in maraging steels. The chemical composition of the steel was smartly designed through thermodynamic calculation and the microstructure of martensitic matrix with nano-sized Mn-rich precipitations was verified by APT and HRTEM analyses. The strength has reached the level of conventional maraging steels, though that of some maraging steels is over 2GPa. Thus the new steel with low cost has much potential for applications. One question is how the impurities in the steel were controlled, as those of the conventional maraging steels?”

Answer: We would like to cordially thank the reviewer for the strong support and important comments. Nickel maraging steels are generally produced by a double vacuum melting procedure to obtain a high standard of cleanliness since small amount of impurities can decrease the toughness. In particular, sulfur and phosphorus should be kept as low as possible (> 100 wt. ppm for aircraft applications). The Fe₁₈Mn₃Ti steel was produced in a similar way in a laboratory scale to ensure similar levels of sulfur and phosphorus. Additionally, the highly segregated edges of the slab were subsequently cut off for better control of the composition. Details about the alloy production are provided in the Methods session:

“An ingot of the Fe-18%Mn-3%Ti (wt.%) alloy used in this work was first synthesized in a vacuum induction furnace and cast as a 60 cm x 40 cm rectangular billet using high-purity metals to obtain a high standard of cleanliness comparable to Nickel maraging steels. The highly segregated edges of the slab were subsequently cut off and the material was hot-rolled from 60 to 6 mm at 1100 °C.”